# A Cyclopentanone Compound Attenuates the Over-Accumulation of Extracellular Matrix and Fibrosis in Diabetic Nephropathy via Downregulating the TGF-β/p38MAPK Axis

**DOI:** 10.3390/biomedicines10123270

**Published:** 2022-12-16

**Authors:** Chunyin Tang, Meng Wang, Jieting Liu, Chunlei Zhang, Luxin Li, Yan Wu, Yanhui Chu, Dan Wu, Haifeng Liu, Xiaohuan Yuan

**Affiliations:** College of Life Science, Mudanjiang Medical University, Mudanjiang 157011, China

**Keywords:** cyclopentanone compound, ECM, renal fibrosis, diabetic nephropathy, curcumin analog, the TGF-β/p38MAPK signaling pathway

## Abstract

Excessive accumulation of the extracellular matrix (ECM) is a crucial pathological process in chronic kidney diseases, such as diabetic nephropathy, etc. The underlying mechanisms of how to decrease ECM deposition to improve diabetic nephropathy remain elusive. The present study investigated whether cyclopentanone compound H8 alleviated ECM over-deposition and fibrosis to prevent and treat diabetic nephropathy. HK-2 cell viability after treatment with H8 was measured by an MTT assay. ECM alterations and renal fibrosis were identified in vitro and in vivo. A pharmacological antagonist was used to detect associations between H8 and the p38 mitogen-activated protein kinase (p38MAPK) signaling pathway. H8 binding was identified through computer simulation methods. Studies conducted on high glucose and transforming growth factor β1 (TGF-β1)-stimulated HK-2 cells revealed that the p38MAPK inhibitor SB 202190 and H8 had similar pharmacological effects. In addition, excessive ECM accumulation and fibrosis in diabetic nephropathy were remarkably improved after H8 administration in vivo and in vitro. Finally, the two molecular docking models further proved that H8 is a specific p38MAPK inhibitor that forms a hydrogen bond with the LYS-53 residue of p38MAPK. The cyclopentanone compound H8 alleviated the over-deposition of ECM and the development of fibrosis in diabetic nephropathy by suppressing the TGF-β/p38MAPK axis.

## 1. Introduction

Generally, as one of the serious renal microvascular complications of diabetes, diabetic nephropathy (DN) is caused by many factors, including over-nutrition, obesity, lifestyle, and genes, which lead to hypertension, continuous urinary albumin, and end-stage renal disease, eventually developing into renal failure [1]. Microscopically, the main structural features of DN are glomerular basement membrane thickening, mesangial structural changes, the destabilization of podocyte foot processes, and the over-accumulation of extracellular matrix (ECM) proteins, which result in renal fibrosis [2]. Thus far, there are no specific treatments to block fibrosis during the process of DN [3]. In addition, the pathogenesis of DN is so intricate that it has not been fully elucidated. Therefore, it is greatly valuable to deeply investigate the pathological characteristics and pathogenesis of diabetic renal fibrosis, as well as continuously identify new drugs for the prevention and treatment of DN.

Transforming growth factor β1 (TGF-β1) is a short peptide that exerts many biological functions, including regulating growth and development, inflammation, and repair [4,5]. A previous study indicated that TGF-β1 activates downstream signaling pathways, such as drosophila mothers against decapentaplegic protein (SMADS) and p38 mitogen-activated protein kinases (p38MAPK), during renal fibrosis [6]. Although the TGF-β1/SMADS signaling pathway has proved to be a dominating signaling pathway for fibrogenesis in kidneys [7,8], new findings reported other fibrotic pathways, such as the extracellular signal-regulated kinases 1 and 2 (ERK1/2) and p38MAPK, may also be involved [9,10]. p38MAPK is a member of the MAPK family, which transmits environmental stimuli to the nucleus and mediates an important intracellular signal. Its activation is closely related to cellular inflammation, apoptosis, and the stress response [11]. Recent studies also manifest that p38MAPK plays an essential part in the development of fibrosis in the renal tissue [12,13]. Additionally, p38MAPK inhibitors, such as SB202190, alleviate renal fibrosis [14]. Consequently, seeking a new, safe, non-toxic, and efficient p38MAPK inhibitor is a novel approach to preventing renal fibrosis in DN.

Curcumin (Cur), as a traditional Chinese medicine, is known for its anti-inflammatory, anti-tumor, anti-oxidation, and blood glucose-reducing properties [15]. Several reports have highlighted the clinical value of Cur. However, the exertion of its curative effect is primarily limited due to its low bioavailability and rapid metabolism [16,17]. Thus, we synthesized a series of Cur analogs. After the screening, we previously reported that the cyclopentanone compound H8 (Figure 1A) showed low toxicity and higher safety compared to Cur, and the bioavailability of H8 was also several times higher than that of Cur. Moreover, it significantly lowered blood glucose and blood lipids in diabetes [18,19]. In addition, we accidentally found that H8 decreased uric acid, urea nitrogen, and creatinine in diabetic rats. However, the study of H8 in DN has not been conducted. Thus, in this study, we aimed to investigate whether the inhibitory effect of H8 on DN is linked to the suppression of the TGF-β1/p38MAPK axis.

## 2. Materials and Methods

### 2.1. Reagents

DMEM-F12 medium (SH30023.01, Hyclone, Logan, UT, USA); penicillin streptomycin solution (SV30010, Hyclone, Logan, UT, USA); FBS (SH30084.03, Hyclone, Logan, UT, USA); HG (D-glucose, Hyclone, Logan, UT, USA); SB202190 (VB2726-10, Viva Bioscience, Shanghai, China); TGF-β1 (AF-100-21C, PeproTech, Rocky Hill, NJ, USA); MTT (GL0247-RXS, BioLite Biotech, Tianjin, China); compound H8 (Mudanjiang Medical University, Mudanjiang, China); CMC-Na (C9481-500G, Sigma, St. Louis, MO, USA); STZ (N407-1G, Amresco, Solon, OH, USA); all the primers (Sangon Biotech, Shanghai, China); anti-Col IV (ab6586, Abcam, Cambridge, UK); anti-FN (ab2413, Abcam, Cambridge, UK); anti-TGF-β1 (ab9758, Abcam, Cambridge, UK); anti-α-SMA (ab5694, Abcam, Cambridge, UK); anti-E-Ca (H-108: sc-7870, Santa Cruz, CA, USA); anti-p38MAPK (#9212, Cell Signaling, Danvers, MA, USA); anti-P-p38MAPK (#9211, Cell Signaling, Danvers, MA, USA); and anti-β-actin (#4967, Cell Signaling, Danvers, MA, USA).

### 2.2. Cell Culture and Treatment

HK-2 cell lines were purchased from the American Type Culture Collection (ATCC, Manassas, VA, USA) and cultured in DMEM-F12 medium containing 100 U/mL penicillin, 100 μg/mL streptomycin, and 10% (*v*/*v*) FBS in an incubator with 5% CO2 (*v*/*v*) at 37 ℃. The cells were grown to approximately 80% confluence. To explore the influence of H8 on high glucose (HG)-stimulated HK-2 cell injury, the cells were simultaneously administered with HG (35 mM D-glucose) and the indicated concentrations of H8 for 48 h and were divided into 5 groups as follows: the normal group; HG group; HG + 5 μM H8 group; HG + 10 μM H8 group; and HG + 15 μM H8 group. To prove that the role of H8 was associated with the p38MAPK signaling pathway, we utilized the p38MAPK inhibitor SB202190 or H8 to treat the HK-2 cells in an HG environment for 48h and set up 4 groups as follows: the normal group; HG group; HG + 10 μM H8 group; and HG + 5 μM SB202190 group. To further define whether the therapeutic effects of H8 were linked to the inactivation of the TGF-β/p38MAPK pathway, the cells were simultaneously administered with TGF-β1 + H8 or TGF-β1 + SB202190 for 24 h. The cells were divided into 4 groups as follows: the normal group; 10 ng/mL TGF-β1 group; 10 ng/mL TGF-β1 + 10 μM H8 group; and 10 ng/mL TGF-β1 + 5 μM SB202190 group.

### 2.3. Cell Viability Assay

Cell viability was measured by an MTT [3-(4,5-dimethylthiazol-2-yl)-2,5-diphenyl tetrazolium bromide] assay. HK-2 cells were seeded onto 96-well plates (Costar, Cambridge, MA, USA) at a density of 4000 cells/mL and were treated with H8 (0, 1, 5, 10, 20, 25, or 50 μM) for 24 h and 48 h. Then, 1 mL DMEM-F12 containing 0.05 mg/mL MTT was added. After 4 h, the media was discarded, and the absorbance was determined at a 570 nm wavelength with a microplate reader (Molecular Devices, Sunnyvale, CA, USA).

### 2.4. Animals and Experimental Protocol

Sixteen male C57BL/KsJ background *db/db* mice and eight male non-diabetic *db/m* mice (22–25 g) were purchased from the Beijing Wei Tong Li Hua experimental animal technology Co., Ltd., Beijing, China, [number of animal license: SCXK (JING) 2019-0009]. The mice were kept in a specific pathogen-free (SPF) environment in the laboratory at the medical research center. The *db/db* mice, with genetic defects of the leptin receptor, have been widely studied as type 2 diabetes and DN models due to their renal structural changes, glomerular enlargement, proteinuria, and mesangial matrix expansion [20,21,22]. Moreover, 8-week-old *db/db* mice exhibit the features of the early stage of DN [23]. Therefore, in this study, 9-week-old mice were divided into 3 groups: the control group (*db/m*; *n* = 8), *db/db* group (*db/db*; *n* = 8), and H8 treatment group (*db/db* + H8; *n* = 8). The mice in the H8 treatment group were intragastrically administered with 5 mg/kg of H8 (dissolved with 1% CMC-Na solution) every day. Meanwhile, the *db/db* group and the control group were also given the same volume of 1% CMC-Na for 8 weeks. At the end of the experiment, urinary and blood samples were collected. All the mice were sacrificed with sevoflurane, and the kidneys were collected in liquid nitrogen and stored at −80 °C for further testing. 

Twenty-four SPF SD rats were obtained from Liaoning Changsheng Biotechnology Co., Ltd., Benxi, China, [number of animal license: SCXK (LIAO) 2015-0001]. The rats were kept under the conditions mentioned above. Six-week-old rats, weighing 170–190 g, were randomly divided into control (Con) (*n*  =  8) or diabetic (*n*  =  16) groups. The diabetic rats were intraperitoneally injected with STZ (25 mg/kg), combined with a high-fat diet. The Con group was intraperitoneally injected with an equal dose of citric acid buffer. Rats with fasting blood glucose levels ≥11.1 mmol/L for 3 consecutive days were considered to have type 2 diabetes. The DN model was established by the diabetic rats. We collected the rat urine via metabolic cages to test volume and protein levels at weeks 4 to 6 after the STZ supplement. When the 24 h urinary protein levels reached ≥ 30 mg/24 h, the diabetic rats were verified as having DN (shown in Appendix A). Then, 16 DN rats were randomly divided into DN + 1% CMC-Na solution (*n* =  8) and DN + H8 (dissolved with 1% CMC-Na solution) (*n* =  8) groups. The DN + H8 group received 3 mg/kg of H8 once a day for 8 weeks via gavage treatment, while the DN and control groups received the same volume of a 1% CMC-Na solution. After 8 weeks of treatment, urine and blood samples were collected. All the rats were sacrificed with sevoflurane, and the kidneys were collected in liquid nitrogen and stored at −80 °C for further testing. 

All the experimental protocols were approved by the committee on the ethics of animal experiments at Mudanjiang Medical University (Mudanjiang, China).

### 2.5. Biochemical Assays

The urine samples were analyzed to detect N-acetyl-β-D-glucosaminidase (NAG), 24 h urinary protein levels, and urine creatinine (Ucr). The blood samples were assessed to detect blood glucose (GLU), uric acid (UA), serum creatinine (Scr), and blood urea nitrogen (BUN). 

### 2.6. Histopathological Staining

The kidney samples fixed in a 10% formalin solution were embedded in paraffin and cut into sections with a thickness of 5 μm. The sections were prepared for hematoxylin-eosin (HE) staining, Sirius Red staining, and Masson staining. HE staining was used to evaluate the pathological changes in the kidney. Masson staining and Sirius Red staining were performed to observe the collagen changes in the kidney.

### 2.7. Real-Time Quantitative PCR (qPCR)

qPCR was executed as we previously reported [18,19]. All the primers are listed in Appendix A.

### 2.8. Western Blot Assay

The Western blot assay was executed according to the methods previously reported [18,19]. The primary antibodies were as follows: Col IV (1:1000); FN (1:1000); TGF-β1 (1:250); α-SMA (1:1000); E-Ca (1:250); p38MAPK (1:1000); P-p38MAPK (1:1000); and β-actin (1:1000).

### 2.9. Immunofluorescence (IF) Staining

For IF staining, the HK-2 cells and renal tissues from the animals were fixed in 4% formaldehyde (pH = 7.5) for 15 min at room temperature. Then, IF was executed according to methods previously described [18,19]. The primary antibodies were FN (1:50) and α-SMA (1:50). Images were captured using a laser scanning confocal microscope (Tokyo, Japan, Olympus: FV1000).

### 2.10. Molecular Docking

AutoDock (version 4.2.6, Scripps Research Institute, San Diego, CA, USA) [24,25] was used to perform molecular docking based on the candidate target for drug screening with the binding position of H8 in p38MAPK’s binding site. The X-ray structure of human p38MAPK (PDB code: 1BL7) was retrieved from the PDB (www.rcsb.org, accessed on 22 August 2022) [26], and the pure p38MAPK protein structure was created with Pymol and Discovery studio software (BIOVIA, 2019 version, Accelrys Software Inc., San Diego, CA, USA).

### 2.11. Data and Statistical Analysis

All the studies were designed to generate groups of equal size using randomization and blinded analyses. The data were assessed using GraphPad Prism software (GraphPad Software Inc., San Diego, CA, USA) with a one-way ANOVA followed by Bartlett’s test for equal variances. Then, the data were analyzed by Tukey’s test to compare all pairs of columns, as appropriate (when there was a significant difference between two groups and there was no variance in homogeneity). All the data are presented as the mean ± standard deviation (S.D.). The difference was considered statistically significant when *p* < 0.05.

## 3. Results

### 3.1. Attenuation of the Over-Amassment of ECM by H8 in the HG-Stimulated HK-2 Cells

HG decreases ECM degradation and increases the epithelial-to-mesenchymal transition (EMT) [27,28]. To identify whether H8 reduced EMT, the optimal concentrations of H8 (5, 10, and 15 μM) were administrated to HG-stimulated HK-2 cells. Firstly, the results of the MTT assay showed that H8 was safe and non-toxic (Figure 1B,C). The qPCR results indicated that H8 reduced the mRNA expressions of α-SMA, FN, Col IV, and TGF-β1 in HG-stimulated HK-2 cells. However, H8 boosted the expression of E-Ca mRNA (Figure 1D). The Western blot analysis demonstrated that the different concentrations of H8 completely inhibited the over-deposition of ECM proteins in the HG-stimulated HK-2 cells (Figure 1E,F). IF further revealed that HG increased the levels of FN in HK-2 cells, and the elevated levels of FN were reduced by H8 (Figure 1G). Meanwhile, the p38MAPK signaling pathway was activated in the HG-stimulated HK-2 cells, but H8 restored these changes.

### 3.2. The Influence of H8 on HG-Stimulated HK-2 Cells May Be Linked to p38MAPK Signaling Pathway

To determine whether the effect of H8 resulted from the inhibition of the p38MAPK signaling pathway, SB202190 was used to treat the HK-2 cells in the HG environment as a positive control. The qPCR results indicated that the H8 treatment significantly reduced the mRNA expressions of α-SMA, FN, Col IV, and TGF-β1 in HG-stimulated HK-2 cells, consistent with the SB202190 group (Figure 2A). On the other hand, H8 and SB202190 also increased the mRNA level of E-Ca. The Western blot results showed that protein changes were similar to the mRNA changes (Figure 2B,C). The effect of H8 on the ECM proteins was consistent with the SB202190 in HG-mediated HK-2 cells. Strikingly, the activation of p38MAPK was significantly inhibited by H8, which suggested that H8 worked like an inhibitor.

### 3.3. H8 Suppressed TGF-β1-Induced p38MAPK Signaling Activation in HK-2 Cells

To further verify the potential molecular mechanism with which H8 acts on the over-accumulation of ECM, the ratio of P-p38MAPK/p38MAPK was examined in TGF-β1-induced HK-2 cells that were administered with H8 or SB202190. H8 and SB202190 effectively improved the variation of the mRNAs compared to the TGF-β1 group (Figure 3A). The results of the Western blot assay showed that H8 and SB202190 had similar pharmacological effects, which effectively reduced the levels of the ECM proteins, increased the expression of E-Ca, and suppressed the activation of the p38MAPK signaling pathway in the TGF-β1-induced HK-2 cells (Figure 3B,C). The IF analysis demonstrated that TGF-β1 upregulated the protein level of FN. However, H8 and SB202190 treatment alleviated the over-accumulation of FN at the same time (Figure 3D).

### 3.4. Two Computational Modelings Showed That H8 Formed a Hydrogen Bond with the LYS-53 Residue of p38MAPK

The binding site of H8 on p38MAPK was predicted by AutoDock software (Figure 4A). The computational modeling results demonstrated that H8 formed a hydrogen bond with the LYS-53 residue of p38MAPK. In addition, H8 inserted into a hydrophobic gap and interacted with TYR-35, VAL-38, ALA-51, LYS-53, GLU-71, LEU-74, LEU-75, ILE-84, THR-106, HIS-107, LEU-108, MET-109, LEU-167, ASP-168, PHE-169, GLY-170, LEU-171, ALA-172, and ARG-173 (Figure 4B,C). In addition, we obtained the docking parameters of H8 with TGF-β1, MAPKs, and AKT (shown in Appendix A).

### 3.5. H8 Treatment Improved the Renal Structure and Function in the db/db Mice

We captured phase-contrast images of *db/db* mice in the different groups to assess tissue pathological changes, which were observed by HE staining, Masson staining, and Sirius Red staining (Figure 5A). The control group presented a normal glomerular structure, but the *db/db* group showed obvious glomerular mesangial expansion (represented by an orange arrow in the HE staining). In addition, renal tubules in the *db/db* group showed more severe renal fibrosis (represented by green arrows in the Masson staining) and the formations of collagen (represented by a black arrow in the Sirius Red staining) than those in the control group. After H8 administration, the aforementioned pathological changes in the kidney were significantly improved. Meanwhile, compared to the *db/db* group, the H8 treatment significantly decreased the levels of BUN, Scr, UA, and GLU in the serum, and improved the levels of 24 h urinary protein, NAG, Ucr, and the albumin/creatinine ratio in the urine (Figure 5C–J). These results indicated that H8 effectively ameliorated renal pathological changes and function in the *db/db* mice.

### 3.6. H8 Reduced Renal Fibrosis and Blocked p38MPAK Signaling Pathway in the db/db Mice

The qPCR revealed that the mRNA expressions of FN, α-SMA, TGF-β, and Col Ⅳ in the kidney tissue of the *db/db* group were abnormally increased compared to the control group, and the H8 treatment vastly decreased these changes. However, the mRNA expression of E-Ca was contrary to other mRNA expressions in the diabetic group and H8 treatment group (Figure 6A). The protein levels of FN, α-SMA, TGF-β1, Col IV, and P-p38MPAK were detected by a Western blot analysis. Compared to the Con group, the expressions of ECM-related proteins in the *db/db* group were dramatically elevated, except for E-Ca. However, these expressions were reversed after H8 administration (Figure 6B,C). Furthermore, H8 also repressed the activation of the p38MAPK signaling pathway in the *db/db* mice. The IF results verified that the protein expressions of α-SMA and FN were increased in the renal tubules of the *db/db* mice (represented by white arrows) and were reversed by the H8 treatment (Figure 6D).

### 3.7. H8 Treatment Improved the Renal Structure and Function in the DN Rats

The structural changes in the glomerular and tubular tissues were evaluated under a microscope after staining (Figure 7A). The rats with DN had an abnormal glomerular architecture with mesangial matrix expansion compared to the Con group (represented by an orange arrow in the HE staining). In addition, compared to the Con group, the Masson staining and Sirius Red staining revealed more blue-stained collagen fibers and collagen fibril precipitation, respectively, in the renal tubules (represented by green arrows in the Masson staining and a black arrow in the Sirius Red staining). However, the H8 treatment improved these renal pathological changes in DN. Moreover, compared to the DN group, the administration of H8 also significantly decreased the levels of BUN, Scr, UA, and GLU in the serum and improved the levels of 24 h urinary protein, NAG, Ucr, and the albumin/creatinine ratio in the urine (Figure 7C–J).

### 3.8. H8 Suppreseed Renal Fibrosis and the p38MPAK Signaling Pathway in the DN Rats

The qPCR results uncovered that the mRNA expressions of FN, α-SMA, TGF-β1, and Col IV in the DN group were notably increased in comparison to the Con group, and the H8 treatment remarkably downregulated the above levels. However, the E-Ca change was contrary to other mRNA expressions (Figure 8A). Compared to the Con group, the expressions of the ECM proteins in the DN group significantly changed, and they were reversed by the H8 treatment (Figure 8B,C). Furthermore, P-p38MPAK was significantly blocked by H8 in the DN rats.

## 4. Discussion

It is well-known that the kidney plays an irreplaceable role in maintaining the balance of waste excretion, hormone production, ionic composition, and acid-base balance in vivo, thus ensuring the coordinated operation between different cells. Once acute and chronic kidney damage occur, a person’s health is seriously threatened [29]. Several studies have shown that HG is a key risk factor for chronic kidney injury, especially in the DN process. HG facilitates the expression of some essential fibrotic factors, further increases the excessive accumulation of ECM proteins, and accelerates the EMT and renal fibrosis [28,30,31,32]. Studies have displayed that HG triggers multiple signaling pathways, such as PI3K, p38MAPK, and SMAD, while activating the crosstalk between ERK and the SMAD pathway [33,34,35].

The activation of these signaling pathways further elevates FN, TGF-β1, and other proteins, and inhibits ECM degradation, which continuously causes direct damage to the innate cells and tissues of the kidney. Not only mesangial cells, but also fibroblasts, podocytes, epithelial cells, and endothelial cells, are severely injured, eventually leading to DN, end-stage renal disease, and chronic renal failure [36,37,38]. It has been reported that the p38MAPK signaling pathway is activated and the expression levels of P-p38MAPK are significantly increased in HK-2 cells treated with 25 mmol/l of HG for 24 h, which results in the excessive accumulation of ECM, followed by cell damage and fibrosis [39]. Furthermore, HG at 40.9 mM increases the production of reactive oxygen species and the expression of TGF-β1, activates p38MAPK, and catalyzes oxidative stress and cell apoptosis in HK-2 cells [40]. Another study confirmed that the p38MAPK and SMADS signaling pathways are activated in HK-2 cells treated with 60 mmol/l of HG for 48 h, which reduces the expression of E-Ca, increases α-SMA, FN, and vimentin protein levels, and aggravates EMT [41]. The extensive loss of E-Ca and the increase of α-SMA are the main reasons that epithelial cells lose their epithelial characteristics and acquire mesenchymal characteristics, which is a key process in renal tubular sclerosis [42,43]. Our data showed that when HK-2 cells were exposed to a 35 mmol/l HG environment for 48 h, Col IV, α-SMA, FN, and TGF-β1 proteins were highly expressed, and E-Ca expression was decreased significantly. Additionally, p38MAPK was activated, and the ratio of P-p38MAPK and p38MAPK protein was dramatically elevated in comparison to the control group, which was consistent with previous reports. However, different concentrations of the compound H8 reversed the expression of the above proteins, improved the over-accumulation of ECM, and inhibited the activation of p38MAPK.

Recently, many studies have reported that inhibiting the p38MAPK signaling pathway is beneficial to alleviate various kidney diseases [12,13,44,45,46]. Studies conducted on kidneys from hyperglycemic mice and HG-stimulated podocytes have shown that p38MAPK is involved in triggering chain reactions associated with hyperglycemia-induced nephrin endocytosis [47]. p38MAPK activation results in a leaky glomerular filter by phosphorylating the c-terminus of nephrin at serine 1146, thus increasing permeability and albuminuria. However, the blockage of p38MAPK decreases hyperglycemia-induced nephrin endocytosis and attenuates albuminuria [47]. Interestingly, p38MAPK activation and Mkk3 protein elevation in diabetic Mkk3 (+/+) *db/db* mice facilitates albuminuria, renal hypertrophy, and podocyte loss. Nevertheless, the levels of P-p38 in diabetic Mkk3 (-/-) *db/db* mice remain low, which provides kidney protection against tubular injury and interstitial fibrosis [48]. This evidence reveals that the inhibition of p38MAPK is indeed beneficial to improve renal function. Notably, in our study, the p38MAPK inhibitor SB202190 group exhibited the same results as H8, which confirmed that H8 alleviated HK-2 cell fibrosis in the HG environment via the p38MAPK signaling pathway.

The TGF-β cytokine family is ubiquitous in most eukaryotes and has multifarious functions that are essential for survival. It exists in all tissues, but is particularly abundant in the kidneys, bones, lungs, and placenta, exerting a key role in growth and development, the inflammatory reaction, wound healing, tissue repair, and host immunity [5]. In general, releasing and activating TGF-β1 stimulates the production of various ECM proteins, accompanied by the inhibition of their degradation. Moreover, excessive TGF-β1 leads to tissue fibrosis and impairs normal organ function in many diseases [4,49]. For the occurrence and development of renal fibrosis, TGF-β1 plays a mediating role, which manifests in three aspects: (i) TGF-β1 directly induces ECM production (such as FN and ɑ-SMA) through Smad3-dependent or non-dependent mechanisms, including MAPKs [50,51]; (ii) TGF-β1 regulates matrix metalloproteinases (MMPs) to increase or decrease ECM [52]; and (iii) TGF-β1 regulates the transition of pericytes cells into myofibroblasts [53]. Some evidence shows that the upregulation of TGF-β1 in the glomerulus or tubulointerstitium or TGF-β1 exogenous additions causes renal fibrosis. However, renal fibrosis is improved by antagonizing TGF-β1 [54,55,56]. Moreover, renal epithelial cells differentiate into mesenchymal phenotypes due to TGF-β1. The main characteristics are cadherin changes, namely E-Ca (epithelium) to N-cadherin (mesenchymal) or a massive loss of E-Ca. However, the expressions of α-SMA, FN, Col I, and Col IV are increased [57]. In the present study, we also found that Col IV, α-SMA, and FN proteins and genes were significantly increased, accompanied by the activation of the p38MAPK signaling pathway, in the TGF-β1-induced HK-2 cells compared to the normal group. Compared to the TGF-β1 group, the H8 and SB202190 treatment groups showed significantly reduced expression levels of Col IV, α-SMA, and FN, as well as the P-p38MAPK/p38MAPK ratio, suggesting that H8 inhibited TGF-β1 and the downstream-related p38MAPK signaling pathways to alleviate diabetic renal fibrosis. In addition, compared to the Con group, the cell morphology of HK-2 exhibited significant changes in the HG and TGF-β1 groups, and the TGF-β1 group showed more obvious morphological changes. However, after the H8 treatment, the aforementioned morphological changes in the HK-2 cells were markedly improved (shown in Appendix A). Two sets of animal experiments, involving *db/db* mice and DN rats, and molecular docking data further confirmed that H8 inhibited the expression of TGF-β1, along with the downstream signal activation of the p38MAPK, thereby reducing the over-deposition of ECM, slowing EMT and relieving DN (Figure 9).

The numerous studies have shown that ECM accumulation occurs in the glomerulus and also in the renal tubules. However, it is mainly deposited in the renal tubules to promote renal dysfunction. Our data showed that H8 improved the ECM accumulation in the renal tubules and inhibited renal tubule fibrosis in DN.

## 5. Conclusions

In summary, on the one hand, H8 directly slowed EMT by reducing a-SMA levels and increasing E-Ca expression. On the other hand, H8 also blocked the TGF-β/p38MAPK signaling pathway to mitigate the excessive deposition of ECM and renal fibrosis, thereby improving DN. These results suggest that the cyclopentanone compound H8 shows promise as a novel agent for preventing DN and developing a targeted therapy for endocrine diseases.

## Figures and Tables

**Figure 1 biomedicines-10-03270-f001:**
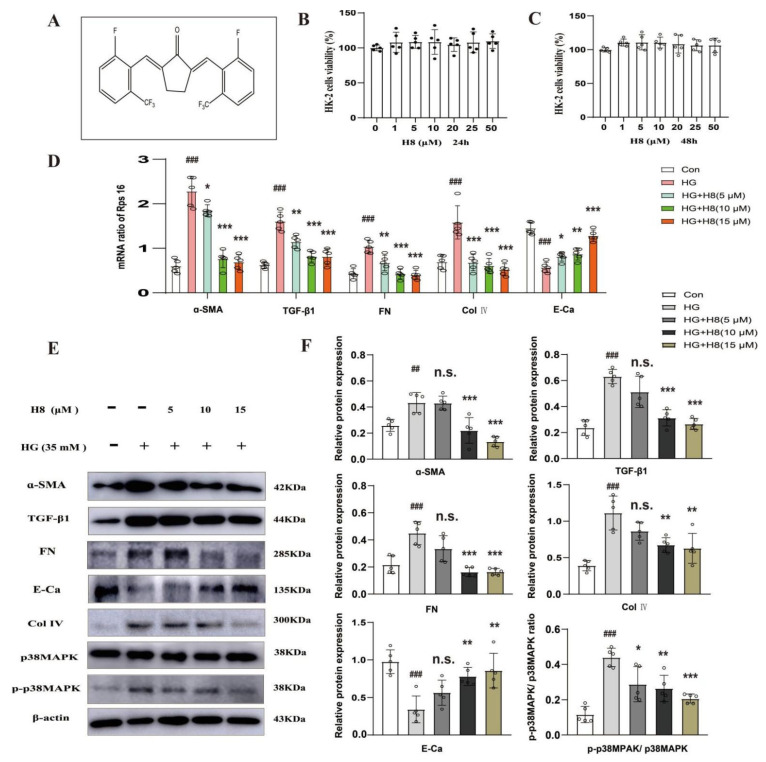
H8 mitigated ECM over-accumulation in HG-stimulated HK-2 cells. (**A**) The H8 chemical structure: (2E, 5E)-2,5-double [(2-fluoro-6-three fluoromethyl) benzyl methylene]-cyclopentone (C_21_H_12_F_8_O; relative molecular mass: 432.31). The effects of different concentrations of H8 on the proliferation of HK-2 cells at the various time points of (**B**) 24 h and (**C**) 48 h. HK-2 cells were co-treated with H8 (5, 10, and 15 μM) and HG (35 mM) for 48 h. (**D**) The mRNA expression levels were examined by qPCR, and the values were normalized to rps16 (α-SMA, TGF-β1, FN, Col IV, and E-Ca). (**E**) Western blot band. (**F**) The protein relative expression levels (α-SMA, TGF-β1, FN, Col IV, E-Ca, and the ratio of P-p38MAPK/p38MAPK). (**G**) The IF was performed using FN antibody, and DAPI was used for staining cell nuclei (scale bar = 50 μm). Data are shown as mean ± S.D. (*n* = 5). ^##^
*p* < 0.01, and ^###^
*p* < 0.001 compared to Con group; * *p* < 0.05, ** *p* < 0.01, and *** *p* < 0.001 compared to HG group; n.s. represents no significant difference in statistics compared to HG group.

**Figure 2 biomedicines-10-03270-f002:**
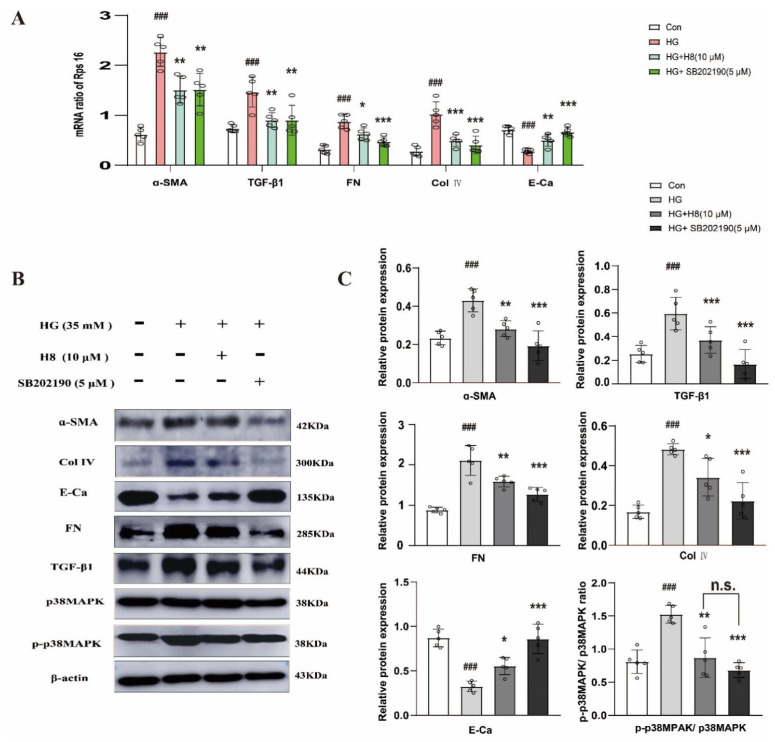
H8 reduced the excessive expressions of ECM in HG-stimulated HK-2 cells, which was related to p38MAPK. (**A**) The mRNA expression levels were examined by qPCR, and the values were normalized to rps16 (α-SMA, TGF-β1, FN, Col IV, and E-Ca). (**B**) Western blot band. (**C**) The protein relative expression levels (α-SMA, TGF-β1, FN, Col IV, E-Ca, and the ratio of P-p38MAPK/p38MAPK). Data are shown as mean ± S.D. (*n* = 5). ^###^
*p* < 0.001 compared to Con group; * *p* < 0.05, ** *p* < 0.01, and *** *p* < 0.001 compared to HG group; n.s. represents no significant difference in statistics.

**Figure 3 biomedicines-10-03270-f003:**
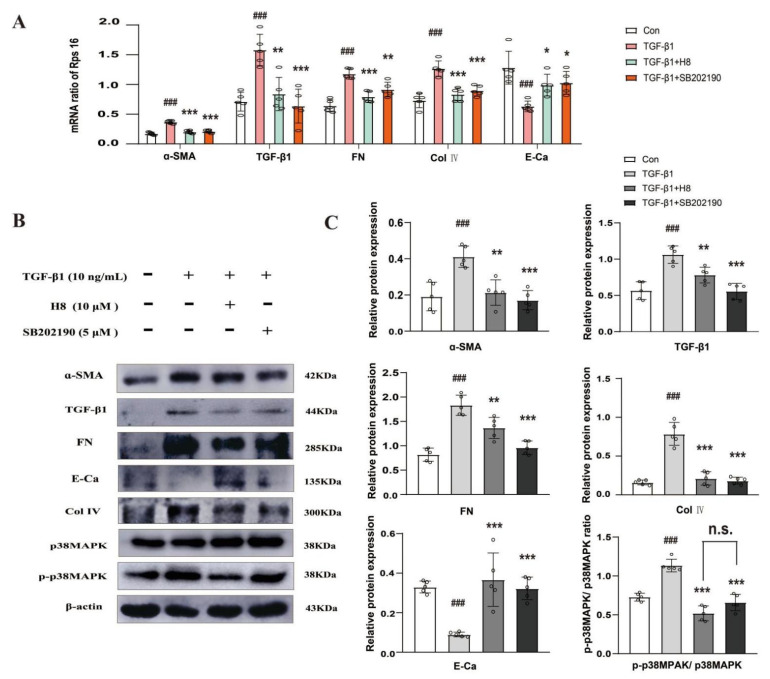
H8 attenuated the over-deposition of ECM and inhibited p38MAPK in the TGF-β1-induced HK-2 cells. (**A**) The mRNA expression levels were examined by qPCR, and the values were normalized to rps16 (α-SMA, TGF-β1, FN, Col IV, and E-Ca). (**B**) Western blot band. (**C**) The protein relative expression levels (a-SMA, TGF-β1, FN, Col IV, E-Ca, and the ratio of P-p38MAPK/p38MAPK). (**D**) The IF was performed using FN antibody, and DAPI was used for staining cell nuclei (scale bar = 50 μm). Data are shown as mean ± S.D. (*n* = 5). ^###^
*p* < 0.001 compared to Con group; * *p* < 0.05, ** *p* < 0.01, and *** *p* < 0.001 compared to TGF-β1 group; n.s. represents no significant difference in statistics.

**Figure 4 biomedicines-10-03270-f004:**
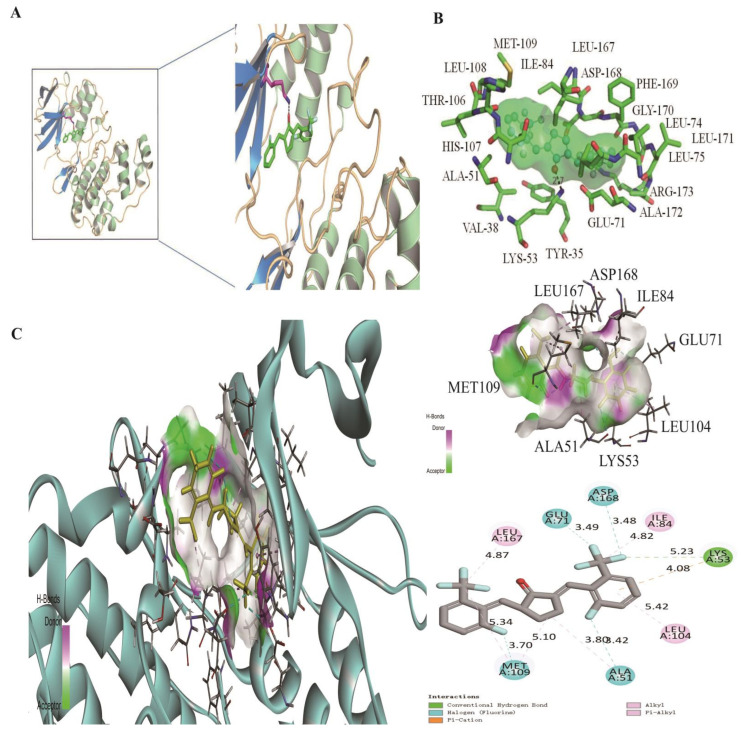
Molecular docking of H8 on the 1BL7 protein. (**A**) Modeling of H8 docking to 1BL7 was simulated with Pymol, and positions of H8 are shown in pink within the active site of 1BL7. (**B**) The green dashed line indicates a possible hydrogen bond between the connected residues and the ligand. (**C**) Modeling of H8 docking to 1BL7 was simulated with Discovery studio software (3D and 2D, San Diego, CA, USA; https://discover.3ds.com/discovery-studio-visualizer-download, accessed on 22 August 2022). All the data showed that H8 could interact with the 1BL7 protein.

**Figure 5 biomedicines-10-03270-f005:**
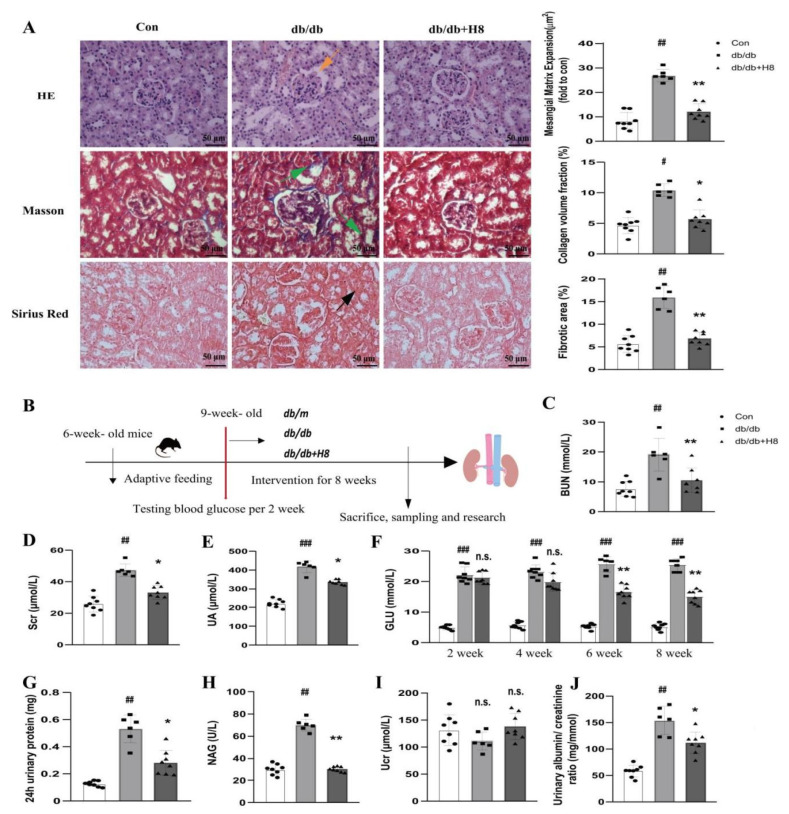
H8 improved the glomerular and tubular pathological changes and renal function in *db/db* mice. (**A**) HE staining, Masson staining, and Sirius Red staining of kidney sections in mice (original magnification 400×; scale bar = 50 µm). Orange arrow represents the glomerular mesangial expansion; green arrows represent accumulation of fibrosis in renal tubule; and black arrow represents formations of collagen in renal tubule. (**B**) The experimental design for the DN treatment in mice. (**C**) BUN (mmol/L). (**D**) Scr (µmol/L). (**E**) UA (µmol/L). (**F**) GLU (mmol/L). (**G**) 24 h urinary protein (mg). (**H**) NAG (U/L). (**I**) Ucr (µmol/L). (**J**) Urinary albumin/creatinine ratio (mg/mmol). Data are shown as mean ± S.D. (*db/m n* = 8, *db/db n* = 6, and *db/db* + H8 *n* = 8). ^#^
*p* < 0.05, ^##^
*p* < 0.01, and ^###^
*p* < 0.001 compared to Con group; * *p* < 0.05, ** *p* < 0.01 compared to *db/db* group; n.s. represents no significant difference in statistics.

**Figure 6 biomedicines-10-03270-f006:**
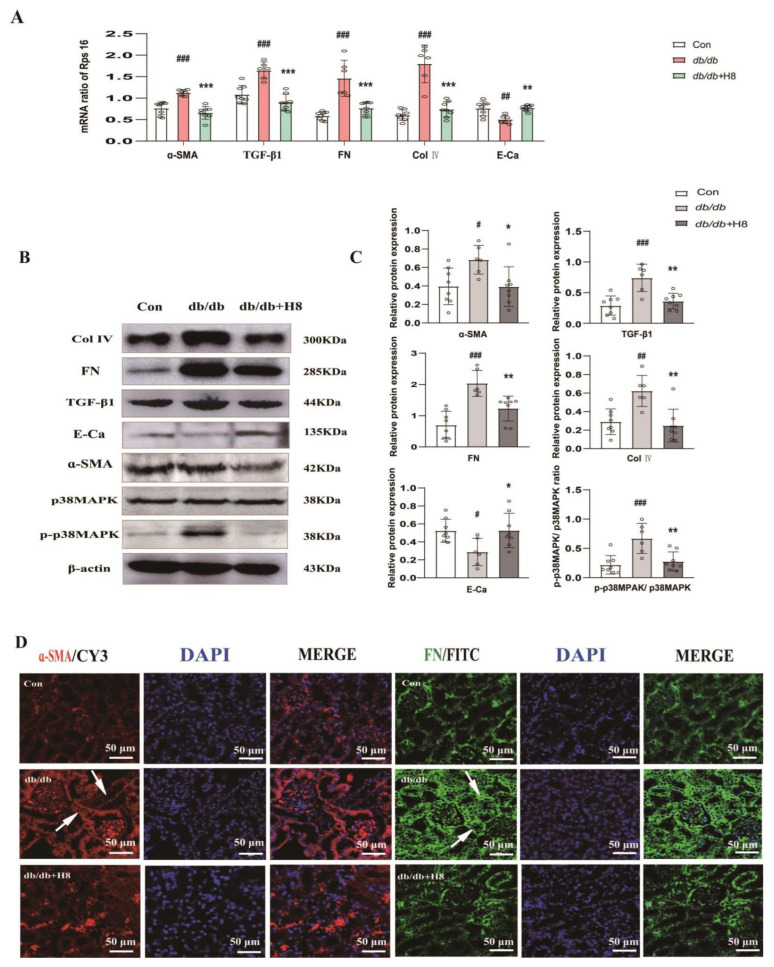
H8 reduced the fibrosis in renal tubule and inhibited p38MPAK signaling pathway in *db/db* mice. (**A**) The mRNA expression levels were examined by qPCR, and the values were normalized to rps16 (α-SMA, TGF-β1, FN, Col IV, and E-Ca). (**B**) Western blot band. (**C**) The protein relative expression levels (α-SMA, TGF-β1, FN, Col IV, E-Ca, and the ratio of P-p38MAPK/p38MAPK). (**D**) IF was performed using α-SMA and FN antibodies, and DAPI was used for staining cell nuclei (scale bar = 50 μm); white arrows represent the protein expressions of α-SMA and FN in renal tubule. Data are shown as mean ± S.D. (*db/m n* = 8, *db/db n* = 6, and *db/db* + H8 *n* = 8). ^#^
*p* < 0.05, ^##^
*p* < 0.01, and ^###^
*p* < 0.001 compared to Con group; * *p* < 0.05, ** *p* < 0.01, and *** *p* < 0.001 compared to *db/db* group.

**Figure 7 biomedicines-10-03270-f007:**
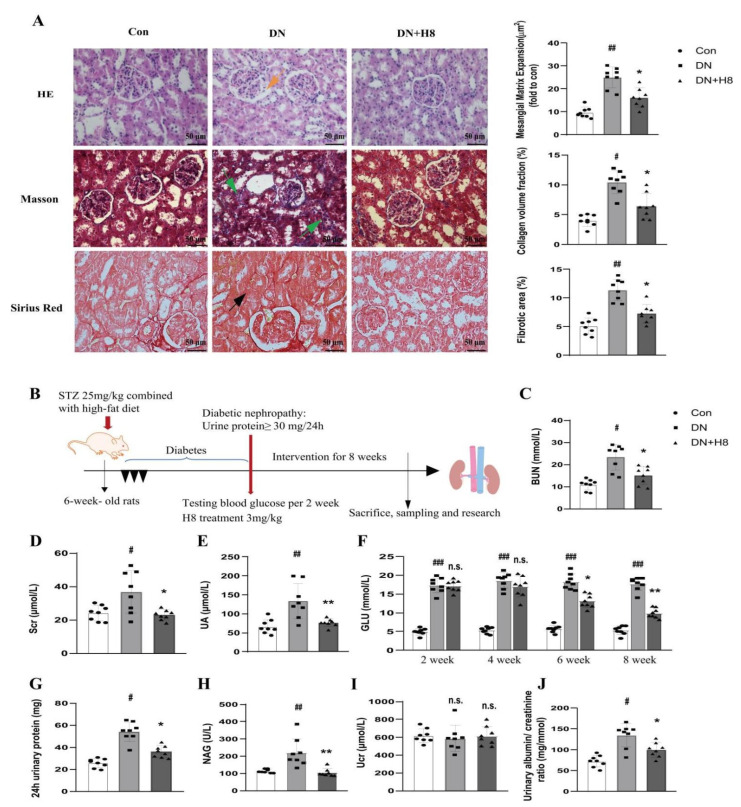
H8 ameliorated renal pathological changes and function in DN rats. (**A**) HE staining, Masson staining, and Sirius Red staining of kidney sections in rats (original magnification 400×; scale bar = 50 µm). Orange arrow represents the glomerular mesangial expansion; green arrows represent accumulation of fibrosis in renal tubule; black arrow represents collagen formations in renal tubule. (**B**) The experimental design for the DN treatment in rats. (**C**) BUN (mmol/L). (**D**) Scr (µmol/L). (**E**) UA (µmol/L). (**F**) GLU (mmol/L). (**G**) 24 h urinary protein (mg). (**H**) NAG (U/L). (**I**) Ucr (µmol/L). (**J**) Urinary albumin/creatinine ratio (mg/mmol). Data are shown as mean ± S.D. (for each group of rats, *n* = 8). ^#^
*p* < 0.05, ^##^
*p* < 0.01, and ^###^
*p* < 0.001 compared to Con group; * *p* < 0.05, ** *p* < 0.01 compared to DN group; n.s. represents no significant difference in statistics.

**Figure 8 biomedicines-10-03270-f008:**
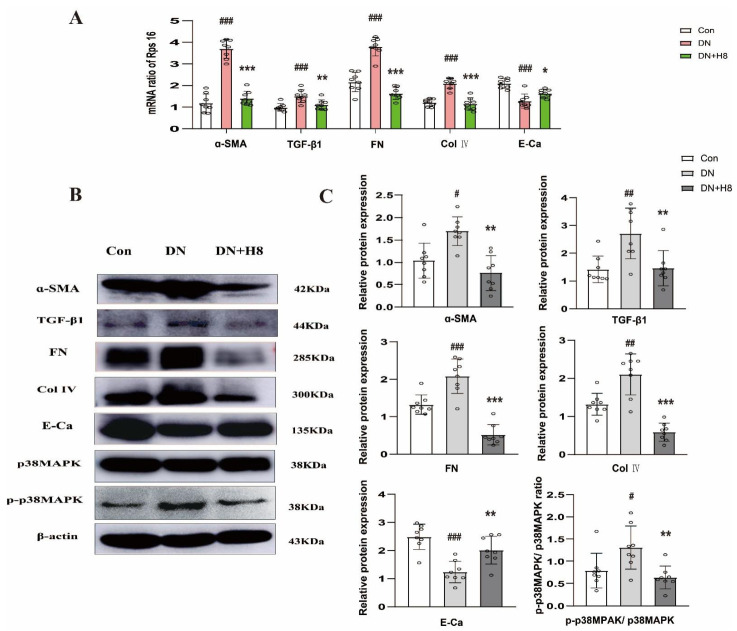
H8 repressed renal fibrosis and p38MPAK signaling pathway in DN rats. (**A**) The mRNA expression levels were examined by qPCR, and the values were normalized to rps16 (α-SMA, TGF-β1, FN, Col IV, and E-Ca). (**B**) Western blot band. (**C**) The protein relative expression levels (α-SMA, TGF-β1, FN, Col IV, E-Ca, and the ratio of P-p38MAPK/p38MAPK). Data are shown as mean ± S.D. (*n* = 8/group). ^#^
*p* < 0.05, ^##^
*p* < 0.01, and ^###^
*p* < 0.001 compared to Con group; * *p* < 0.05, ** *p* < 0.01, and *** *p* < 0.001 compared to DN group.

**Figure 9 biomedicines-10-03270-f009:**
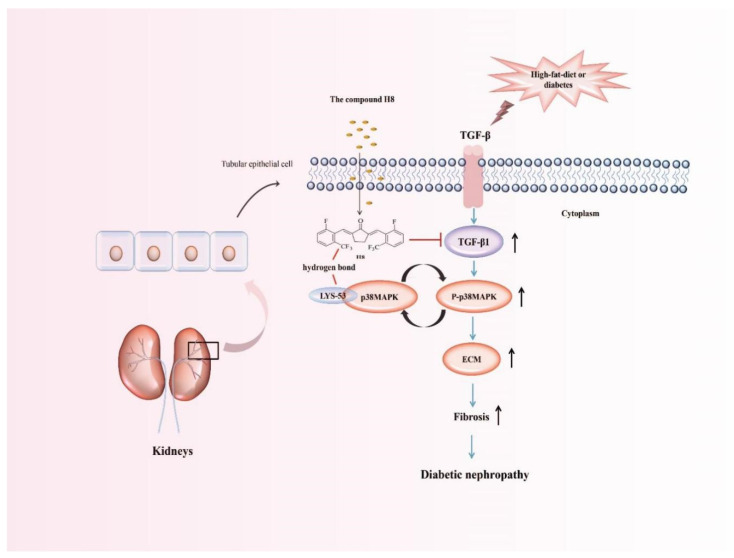
The cyclopentanone compound H8 improved DN via suppressing TGF-β/p38MAPK axis in vitro and in vivo.

## Data Availability

The data presented in this study are available in the Appendix A.

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
