# Peer review of "A Cyclopentanone Compound Attenuates the Over-Accumulation of Extracellular Matrix and Fibrosis in Diabetic Nephropathy via Downregulating the TGF-β/p38MAPK Axis"

_biomedicines, 2022, doi:10.3390/biomedicines10123270_

Round 1

Reviewer 1 Report (Previous Reviewer 2)

The manuscript has been improved. Two remaining items:

1. Please define what the arrows are meant to be showing in Figures 5 and 7 in the legend, or remove them.

2. Please add the docking data given in the responses to reviewers, along with the accompanying explanation, in the figures or as a supplementary figure.

Author Response

Reviewer 2 Report (Previous Reviewer 1)

The authors carefully addressed to almost all the shortcomings designated by the reviewers, though my recommendation to issue a section with all abbreviations was not taken into account, unfortunately. Moreover, I have noticed that abbreviation EMT was not deciphered. As for me, the sense is evident, but this is not a well-known term that does not require deciphering, such as ATP or DNA/

And finally, there is a semantic or even factual inaccuracy on Line 385:  “TGF-b1 regulated the transition of pericytes cells (including epithelial cells and endothelial cells) into myofibroblasts”.  It is a well-known fact that pericytes does not include epithelial and endothelial cells; it is a distinct cell type.

Author Response

Reviewer 3 Report (New Reviewer)

In this study, the cyclopentanone compound H8 was shown to decrease over-deposition of ECM and the development of fibrosis in diabetic nephropathy by inhibiting the TGF-β/p38MAPK. Below is my comments.

1.     Change “… we will investigate” to “… we investigated” (p.1 line 18).

2.     Provide a full name of SMADS (p.2 line 48). Abbreviated words should be written by their full names at first throughout the manuscript.

3.     How did you determine the dose of H8 as 5 mg/kg in db/db mice or 3 mg/kg in rats? How did you determine the duration of H8 treatment as 8 weeks in animals?

4.     Provide company names and catalog numbers for products used in the study.

5.     In Fig. 1G and Fig. 3D, the images of FN immunofluorescence staining were too small, increase resolution and size of high magnification. Is there any morphological difference in Con versus HG-stimulated or TGF-β1-treated HK cells?

6.     What is the rps16? Provide its full name.

7.     Authors showed an upregulation of α-SMA, TGF-β1, FN, Col IV and E-Ca in HK-2, proximal tubular cells to show over-accumulation of ECM in vitro (Fig. 1-3). However, authors analyzed the glomerular accumulation of ECM in vivo as shown in Fig. 5A and Fig. 6D. Why did you use HK2 cells instead glomerular cells to show the changes in expression and secretion of fibrogenic factors (α-SMA, TGF-β1, FN, Col IV)? Please discuss this in the revised manuscript.

8.     Why did you perform two sets of animal experiments involving db/db mice and DN rats? Explain the pathological difference between these two DN models.

9.     Do authors think that H8 has dual function as 1) p38MAPK inhibitor similar to SB 202190 and 2) TGF-β1 blocker as shown in Fig. 9? However, Fig. 4 only support a molecular docking of H8 on the 1BL7 protein, p38MAPK. Did you perform a computational modeling of H8 with TGF-β1?

10.  The title of current paper is not clearly written. “TGF-β-related signaling pathway” is vague. Change the current title by using exact scientific words to describe your finding.

11.  In materials and methods, the number of animals used for each group is 8, however, you showed only one sample from total 8, which is not the representative for the group. Please show at least 3 samples for your WB to conform the results.

12.  Original images mean the whole blots not chopped for the expected band only. Please show whole blots for WB. Perform again your WB analysis with 3 samples per group and show whole blots.

Round 2

Reviewer 3 Report (New Reviewer)

Please include cell morphology data in the supplementary figures. Please include at least 2-3 samples per group for WB presentation in the future. Your manuscript should be reviewed by a native speaker before publication.

Author Response

This manuscript is a resubmission of an earlier submission. The following is a list of the peer review reports and author responses from that submission.

Round 1

Reviewer 1 Report

Diabetic nephropathy (DN) is caused by many factors, which eventually lead to renal failure. The main structural features of DN are glomerular basement membrane thickening and the over-accumulation of extracellular matrix (ECM) proteins, which result in renal fibrosis. There are no specific medicines to block fibrosis during process of DN. Earlier, the authors have synthesized a series of curcumin analogs, and one of them, cyclopentanone compound H8, was previously reported to be of low toxicity and safety compared with curcumin; moreover, the bioavailability of H8 was several times higher than curcumin. In this study, the authors investigated whether the inhibitory effect of H8 on DN is linked with suppression of TGF-β1/p38MAPK axis. H8 directly slowed EMT by reducing alpha-SMA levels and increasing E-Cadherin expression; also, H8 blocked the TGF-β/p38MAPK signaling pathway to mitigate excessive deposition of ECM and renal fibrosis, thereby improving DN. These results suggest that curcumin analog H8 shows promise as a novel agent for preventing DN and developing targeted therapy of endocrine diseases.

Goal-setting is clearly expressed in this article; a wide range of modern methods is used to solve the tasks.

Of the shortcomings, typos should be noted (for example, “paly” vs play, line 312); concentration of streptomycin is 100 um/ml, not 100 mg/ml (line 75); abbreviation HG is not deciphered at the first mentioning (line 77); also, it is recommended to issue a section with all abbreviations.  

Reviewer 2 Report

The authors describe a cyclopentanone compound, H8, derived from curcumin, which inhibits high glucose-induced p38 activation and matrix production by a the proximal tubular cell line HK-2. It also inhibits features of diabetic kidney disease (DKD) in two models of type 2 diabetes, mouse and rat, associated with normalization of blood glucose.

Overall, the presentation of both cell culture data and two different animal models is a strength.

Comments:
Overall the western blot quality is poor. The images are often overexposed, such that it is difficult to see a difference between some of the groups as suggested by the graphs. Examples include, but are not limited to, Fig 1E for col IV, Fig 6B for col IV and 7B for TGFβ1 and p-p38. Figure 1E, I don’t see an αSMA response to HG.

In Figure 2B, the blots don’t show a reduction by H8 in the various proteins assessed, despite what is shown in the graphs. Better blots need to be presented. Certainly the p38 inhibitor seems to be more effective than H8. This is similar for Fig 3B.  Here also, why does SB202190 not decrease p38 phosphorylation in the blot?

How are the data analysed for the graphs? The y axis states “relative protein expression”, but the control average is at significantly less than 1 (eg 0.2). What are the data relative to?

The docking data are interesting. How specific is H8 for p38? Does it affect other MAPKs, Akt, NFkB etc downstream of TGFβ1?

The main issue with the animal data is that the animals are no longer diabetic. Thus, this paper shows that H8 is an effective anti-diabetic agent, and the authors have already shown this in ref 14. If glucose is controlled to essentially normal levels, there will be no diabetes pathology, including the increase in p38 phosphorylation. This is not discussed or acknowledged anywhere. The title is thus misleading in addition to the presentation of results and conclusions.

The pathologic description of the animal data requires quantification for Masson and Sirius red. If inflammatory cell infiltration is pointed out, please use appropriate stains for infiltrates (eg macrophages, T cells) and quantify. For basement membrane, this cannot be assessed on Sirius red, or any other light microscopy in general. EM needs to be done. Please present EM or remove this statement.

Also, I don’t understand what kidney index is measuring (Figure 5F, J).

The discussion needs to include a review of the literature on any studies in which p38 is inhibited and DKD phenotype assessed. For example: doi: 10.1007/s00109-022-02184-5 (Inhibition of p38 MAPK decreases hyperglycemia-induced nephrin endocytosis and attenuates albuminuria); DOI: 10.1007/s00125-008-1215-5 (The reduced levels of p38 MAPK signalling in the diabetic kidneys of Mkk3 ( -/- ) db/db mice were associated with improved DKD phenotype, without effect on hyperglycemia).

On a quick scan, the following gels appear to be different from the raw data: Figure 1 p38, Figure 2 αSMA, col IV, Figure 6 p38 (why is the background so dark here - bands are much easier to see in the raw data), Figure 7 αSMA (this one appears flipped).

Minor comments:
- What does the TGFβ1 antibody from Abcam detect - is this LAP? The indicated size is high for the active cytokine.
- The manuscript could use a read for English - some use of terminology and expression could be improved for readability.
- HG in the graphs should read mM not μM
- Is the kidney weight adjusted for body weight (Figure 5 E, I) as stated in methods? If so please indicate this.
- First paragraph of the discussion - replace “acidity” with “acid-base balance” and remove “osmotic pressure”. HG is specific to DKD, thus lines 316-317 should be reworded.

Reviewer 3 Report

Line 105: was the treatment daily for 8 weeks or Monday to Friday? When was the urine collected? At the end of the experiment?

Line 116: How were the rats fasting? For how long? Overnight?  What does it mean three times in the same day? What were the intervals between the blood collections? How were they taken? What was the time point of sacrifice after the onset of the injury?

Line 150: Microscope brand?

Line 161: Data ‘were’ analyzed.

Line 169: administered ‘to’ the HG-induced cells. Of note: ‘stimulated cells is probably a better expression than ‘induced’.

Line 191: ‘results’ instead of ‘is resulted’

Figure 5B: In the Mason trichrome staining, the left arrow points to vascular collagen fibres, not peritubular collagen deposition. The lowest panel in Figure 5B: Are the authors sure this is a Serius red staining? The colouration does not seem to be in line with the typical appearance of this stain. For example, just compare it to the lowest panel in Figure 5A; there is a significant difference between the two. What is ‘the kidney index of mice’? The arrows are not explained in the figure legends. What sort of ‘glomerular pathological changes’ did the authors observe and how were these determined/quantified?

The standard abbreviation for microalbuminuria is ACR (albumin to creatinine ratio). ‘MAU’ is not common, e.g. this reviewer has never heard it. Were ‘MAU’ and NAG normalized to urinary creatinine? 

Line 276: ‘reduced’ better than ‘retarded’. ‘Blocked’ instead of ‘bloked’.

Figure 6D: The immunofluorescence is inaccurate and very likely just reflects the autofluorescence of the renal tissue. aSMA is either present in the vascular beds, tubulointerstitium or glomerular ECM. What the db/db group shows is autofluorescence of tubules and likely erythrocytes in the tubules. The same applies to fibronectin staining.